# Diet and chemical defenses of the Sonoran Desert toad

**Marina D. Luccioni**[1][⊛]*, **Jules T. Wyman**[1][⊛]*, **Edgard O. Espinoza**[2], **Lauren A. O'Connell**[1]

**1** Department of Biology, Stanford University, Stanford, California, United States of America, **2** Clark R. Bavin National Fish and Wildlife Forensics Laboratory, Ashland, Oregon, United States of America

⊛ These authors contributed equally to this work
* m.luccioni@stanford.edu (MDL); jules.wyman@gmail.com (JTW)

## Abstract

The Sonoran Desert toad (*Incilius alvarius*) is the only animal known to secrete the psychedelic compound 5-MeO-DMT as a chemical defense, but the source of 5-MeO-DMT in *I. alvarius* remains unknown. Some amphibians produce chemical defenses endogenously or through symbiotic interactions, while others acquire them from specialized diets. In this study we analyzed toxin gland secretions and diet profiles from wild *I. alvarius* and sympatric anurans from native and urban habitats around Tucson, Arizona to explore possible links between diet and 5-MeO-DMT production. All *I. alvarius* secreted high concentrations of 5-MeO-DMT, whereas other sympatric toads did not. The diet of *I. alvarius* was similar to that of sympatric anurans, indicating that *I. alvarius* does not exhibit relative dietary specialization. We found slight dietary differences between *I. alvarius* in native and urbanized habitats. Taken together, these lines of evidence suggest that diet is not directly linked to 5-MeO-DMT production and support the alternative hypotheses that *I. alvarius* synthesizes 5-MeO-DMT endogenously or via a microbial symbiont.

## Introduction

Secondary metabolites are bioactive substances that do not directly contribute to an organism's growth and are highly variable in both composition and mechanism of action [1,2]. These small molecules are the targets of intensive biomedical research to treat conditions from chronic pain to cancer [3–5]. Many secondary metabolites play important roles in ecological processes such as predation and defense [6–9]. Animals are thought to acquire chemical defenses through three main routes [10]: (1) taking elemental building blocks from the environment and producing toxins endogenously (e.g., many venomous snakes produce Phospholipase A2 [11]), (2) via symbionts (e.g., rough-skinned newts [*Taricha granulosa*] secrete tetrodotoxin produced by a bacteria living on their skin [12]), and (3) sequestering chemicals from

**Data availability statement:** All code and associated data, and all prey images, are available from the Dryad Digital Repository (DOI: 10.5061/dryad.76hdr7t71). Sequence data are available from GenBank (accession numbers PV892399-892433). All other data from this study are available within the manuscript and its Supporting information files.

**Funding:** LAO received funding from the New York Stem Cell Foundation (https://nyscf.org/) for this work. LAO is a New York Stem Cell Foundation – Robertson Investigator. The New York Stem Cell Foundation did not play any role in the study design, data collection and analysis, decision to publish, or preparation of this manuscript.

**Competing interests:** The authors have declared that no competing interests exist.

dietary organisms (e.g., dendrobatid poison frogs sequester alkaloids from invertebrate prey [13,14]). While the sources of some natural products used for chemical defense have been established, the origins of psychedelic compounds are comparatively understudied.

The Sonoran Desert toad (*Incilius alvarius*) secretes a mix of indolealkylamine secondary metabolites, including the potent hallucinogenic compound 5-Methoxy-N,N-dimethyltryptamine (5-MeO-DMT) [15–17]. These are large toads native to the United States [18], known to eat a variety of invertebrates, and occasionally vertebrates [19,20]. *Incilius alvarius* is found primarily in the Sonoran Desert region of Southern Arizona, USA and Sonora, Mexico. Like other bufonids, *I. alvarius* secretes a mix of toxins from its specialized parotoid and other glands when threatened. Secretions of many species in the family Bufonidae contain potent and pharmacologically active compounds, including indolealkylamines such as bufotenine, a serotonin agonist and metabolite of 5-MeO-DMT [21]. However, unlike related species, *I. alvarius* secretions contain high concentrations of 5-MeO-DMT. In fact, *I. alvarius* is the only animal currently known to secrete 5-MeO-DMT, though it is also found in several plants [22]. 5-MeO-DMT causes marked behavioral and psychological effects in humans, which led to its popularization as a recreational drug in the 1980s [23]. In recent decades, there has been significant growth in use and demand for *I. alvarius* secretions, resulting in widespread illegal toad poaching [24].

The source of 5-MeO-DMT in *I. alvarius* is unknown. Erspamer et al. [17], who originally described 5-MeO-DMT in the skin of *I. alvarius*, suggested that *I. alvarius* produces 5-MeO-DMT endogenously from 5-Hydroxy-dimethyltryptamine (bufotenine) with a 5-Hydroxyindole-O-methyltransferase, a theory adopted by other authors since [25]. However, the existence of this enzyme in toad parotoid glands has not been confirmed. Meanwhile, multiple lines of evidence suggest that the diet of *I. alvarius* might be linked to their secretion of 5-MeO-DMT and other compounds used in defense. Savitzky et al. [26] examined patterns of defensive sequestration across tetrapods and predict that an untested species will sequester defensive compounds if it is (1) from a known sequestering taxon, (2) consumes toxic prey items, and (3) uses passive defense. *Incilius alvarius* fits all these criteria as a member of the family Bufonidae [27,28] which consumes potentially toxic ants and beetles [19,29] and displays a sitting 'puff' behavior when approached by predators [16].

In this study, we investigate whether *I. alvarius* specializes on chemically defended prey and if such a specialization correlates with the presence of 5-MeO-DMT in *I. alvarius* secretions. Relatively few studies [17,30] profile the chemicals secreted by *I. alvarius*, and dietary studies conducted over 60 years ago involved small sample sizes (n ≤ 6) [19,28,31,32]. We analyzed the stomach and fecal contents and tested for 5-MeO-DMT and other secondary metabolites in the secretions of *I. alvarius* from several localities comprising multiple habitats in the vicinity of Tucson, Arizona in the United States. We also sampled diet from three sympatric anuran species (including two bufonids) and examined compounds in the secretions from both sympatric bufonids for comparison. We predicted that if *I. alvarius* sequesters 5-MeO-DMT from a dietary source, we would see a relationship between diet composition and

5-MeO-DMT load, and/or specialization in arthropod taxa that could produce 5-MeO-DMT relative to sympatric anurans not known to produce 5-MeO-DMT. To our knowledge, this is the first study to investigate the relationship of diet and 5-MeO-DMT in *I. alvarius*.

## Materials and methods

### Field sampling

From August-September 2020, we sampled four anuran species–Sonoran Desert toads (Bufonidae: *Incilius alvarius*, N = 29; average mass 191.2g, range 0.5g-400.0g), red-spotted toads (Bufonidae: *Anaxyrus punctatus*, N = 7, average mass 16.3g, range 5.6g–28.0g), Great Plains toads (Bufonidae: *Anaxyrus cognatus*, N = 4, average mass 71.8g, range 41.0–88.0g) and Couch's spadefoots (Scaphiopodidae: *Scaphiopus couchii*, N = 10, average mass 27.7g, range 20.5–38.5g)–across five sites near Tucson, Arizona. As habitat influences prey availability, we characterized the biotic community [33] at each site and identified whether the habitat was predominantly urbanized (N = 2) or native (N = 3) (Table 1). We also classified habitat as either riparian (at a river or arroyo), semi-riparian (at an ephemeral pool), or non-riparian (away from any surface water). As illegal collection of *I. alvarius* to harvest 5-MeO-DMT may present a significant threat to the species [23], we do not disclose exact sampling localities here, but this information is available upon request from the corresponding authors.

To find our study species, we walked with flashlights through suitable habitats on rainy or humid nights. We sampled individuals at various life stages, from juveniles to large adults. We collected anurans using nitrile gloves and stored them individually in clean plastic buckets with air and local substrate. We photographed, weighed, and measured lengths of each anuran before collecting secretions and recent prey items. While we recorded sex based on visual markings for most *S. couchii* and several *Anaxyrus* spp., we were unable to visually sex *I. alvarius* and other individuals. We did not collect secretions from Couch's spadefoots because they lack prominent parotoid glands. We did not weigh or measure three *I. alvarius*, five *S. couchii* and two *A. punctatus* due to field constraints.

To collect recent prey items, we used a nonlethal procedure to flush the anurans' stomachs and collected fecal samples from buckets in which the toads were held. We followed methods described by Solé et al. [34] to prepare a tube-syringe setup with distilled water. We used a mini spatula to open each individual's mouth, inserted a soft tube through the esophagus into the stomach, emptied the syringe, and then collected flushed stomach contents with a strainer. We stored flushed stomach contents and fecal samples separately in 1 ml of 100% ethanol. We flushed the stomachs of all 50 individuals sampled and collected fecal samples for 22 of the 50 individuals comprising all four study species (for more details

**Table 1. 2020 Pima County Anuran Sampling Localities. Ia =** *Incilius alvarius*, **Ac =** *Anaxyrus cognatus*, **Ap =** *Anaxyrus punctatus*, **Sc =** *Scaphiopus couchii*. **Biotic community classifications are based on Brown [33].**

| Locality Designation | General Location | Habitat (Biotic Community) | Elevation (m) | Ia | Ac | Ap | Sc |
|---|---|---|---|---|---|---|---|
| P | West Tucson | Urbanized Riparian (Sonoran Desertscrub: Arizona Upland Subdivision) | 716 | 11 | 1 | 2 | 3 |
| R | West Tucson | Urbanized Semi-Riparian (Sonoran Desertscrub: Arizona Upland Subdivision) | 726 | 6 | 3 | 3 | 3 |
| A | East Green Valley | Native Non-Riparian (Chihuahuan Desertscrub) | 958 | 1 | 0 | 0 | 0 |
| W | Southeast Tucson Basin | Native Riparian (Sonoran Desertscrub: Arizona Upland Subdivision) | 901 | 1 | 0 | 0 | 4 |
| N | Lower Rincon/Catalina Mountains | Native Riparian (Sonoran Desertscrub: Arizona Upland Subdivision) | 1011 | 10 | 0 | 2 | 0 |

see S1 Table). Of the 50 sampled anurans, eight had empty stomachs and did not produce fecal samples, and a further six samples were incorrectly stored, rendering them unsuitable for visual or molecular identification.

We recorded the exact location of each anuran and released them at their capture sites after processing. We decontaminated all gear between field sites using a 5–10% bleach solution [35]. All fieldwork was completed under Arizona Game and Fish Department Scientific Collecting Licenses (#SP404533 and #SP407194). All field sites were on public land, and access to these sites was approved by the Arizona Game and Fish Department and stipulated in our licenses. No toads were sacrificed or otherwise euthanized for this study, as non-lethal procedures were developed for stomach flushing and collecting secretions. Our procedures did not use anesthesia but care was taken to use materials (e.g., silicon tubing) and actions (e.g., gentle rubbing) to minimize animal discomfort. All animal work was approved by the Institutional Animal Care and Use Committee of Stanford University (Protocol #33849).

## Mass spectrometry

We analyzed secretions from parotoid and/or leg glands from 26 *I. alvarius*, six *A. punctatus* and three *A. cognatus.* We gently rubbed the parotoid glands to manually induce secretions in the field and collected secretions using kimwipes [36]. We stored secretion-soaked kimwipes in glass vials containing 2–5 ml of 100% methanol at −20°C until further processing. When individual toads produced more secretions than could be held by one kimwipe, or in cases of accidental contamination with blood from the toxin extraction, we split samples across glass vials. For one *I. alvarius* we used a separate kimwipe for the dorsal area, ventral area and combined parotoid and leg glands. For one further *I. alvarius* we swabbed the leg and parotoid glands separately. For full details of sampling protocols for each toad, see S2 Table.

Chemical analysis was conducted using a Direct Analysis in Real Time ionization coupled to a time-of-flight mass spectrometer (DART-TOFMS) (AccuTOF-DART by JEOL, USA, Inc., Peabody MA USA). This process consisted of dipping a glass capillary melting point tube into the methanol solvent within the sample vials and holding the glass tube in the helium DART gas stream, which was set to 350 °C. Mass spectra were extracted from the mass-calibrated centroided files and the resulting high resolution ions (m/z) were compared against a database which included molecular components described in the literature. Presumptive assignments were made of compounds that had a threshold greater than 5% of the base peak and masses within 5 mmu of the nominal molecular weight.

To characterize chemical profiles of *I. alvarius* and sympatric species, we searched within each sample for the presence of several tryptamines and derivatives recorded by previous authors [29,37–39]. We then compared the presence and absence of these compounds across individuals species and sites (see S3 Table).

## Stomach sample visual identification

We processed stomach and fecal contents by sorting specimens in petri dishes. We photographed dietary items with a Lumenera Infinity 2 camera mounted on an Olympus dissecting microscope (SZ40) and returned the samples to 100% ethanol at 4ºC for storage. We uploaded many of our photos to the online citizen science platform BugGuide [40] for identification assistance. We visually identified nearly all arthropods to order, and further identified many to genus or species, with input from numerous expert contributors (see Acknowledgements). We additionally classified arthropods as native or introduced species. All photos of prey items are available on Dryad (https://doi.org/10.5061/dryad.76hdr7t71).

## Stomach sample molecular identification

We isolated DNA from arthropods using the DNEasy Blood & Tissue kit (Qiagen, Hilden, Germany) with adjustments to suit arthropod tissues. We cut the arthropods into ~0.3 cm³ (lentil-sized) chunks and crushed them with a plastic pestle in a microcentrifuge tube, then incubated samples at 56°C overnight in Proteinase-K and lysis buffer 'ATL' from the DNEasy Blood & Tissue kit. We extracted genomic DNA according to the manufacturer's instructions and stored it at -4°C for downstream amplification. We used PCR to amplify a segment of the cytochrome oxidase 1 (*CO1* or *cox1*) gene from

mitochondrial DNA, a standard marker for DNA barcoding [41–43]. We amplified DNA using general arthropod primers mlCOIintF (5'- GGWACWGGWTGAACWGTWTAYCCYCC) and LOBO-R1 (5'-TAAACYTCWGGRTGWCCRAARAAYCA) [44,45]. For all reactions, we used 1.2 µL of each primer (10 µM) and 21.6 µL of 2X Phusion High-Fidelity PCR Master Mix with GC Buffer (New England Biolabs, Ipswich, MA, USA) in a total reaction volume of 40 µL. We used the following PCR program to amplify *CO1*: 98°C for 60s; 5 cycles of 98°C for 10 s, 48°C for 120 s, 72°C for 1 min; 30 rounds of 98°C for 10 s, 54°C for 120 s, and 72°C for 1 min; and a single incubation of 72°C for 5 min. We ran PCR reactions on a 1% SyberSafe/agarose gel (Life Technologies). We purified successful reactions with a single band of the expected size with the E.Z.N.A. Cycle Pure Kit (Omega Bio-Tek, Norcross, GA, USA). Purified PCR products were Sanger sequenced by GeneWiz Inc. (Cambridge, MA, USA) and we uploaded sequences to GenBank (accession numbers PV892399–892433).

We used DNA barcode sequences to identify the prey items recovered from stomach contents. Barcode sequences were imported into Geneious Prime (v 2025.1.3) for quality trimming and alignment of forward and reverse sequencing reactions. We used nucleotide BLAST from the NCBI Genbank nr database to identify arthropod sequences to the species level. For genomic data we considered results that yielded greater than 97.5% sequence similarity as sufficient to assign genus or species, a more conservative threshold than previous studies [41,46] to better align with our visual identification results. For specimens with less than 97.5% Genbank similarity, we assigned an order or family only when the top BLAST result matched that of visual identification.

## Prey item identification summary

We recorded 280 dietary arthropods for 37 of the 50 anurans sampled, as well as many rocks, sand, and plant materials. Following DNA extraction, we confirmed 166 arthropod samples contained DNA of suitable quality and quantity for downstream molecular work. Samples from multiple (mixed) animals or where PCR amplification failed to produce a band when visualized on agarose gel were excluded from downstream analysis, resulting in 86 samples suitable for Sanger sequencing. We successfully assigned 30 (34.9%) sequenced samples with a taxonomic ID using this method. Across molecular and visual methods, we identified 236 (84.3%) samples to family, 58 (20.7%) samples to genus, and 35 (12.5%) samples to species. Of our genus and species-level identifications, 8 (13.8%) were the result of Sanger sequencing only, 43 (74.1%) were the result of visual examination only and 7 (12.1%) were the result of a consensus identification between visual and molecular methods. In only one case did our molecular and visual methods result in conflicting identifications (*Pogonomyrmex barbatus* vs. *P. rugosus*, see Discussion).

## Data analysis and visualization

We performed data analyses using RStudio version "Ghost Orchid" (2021.09.0 + 351.pro6) and R version 4.1.2. For all data analyses, prey items from stomach and fecal samples were treated together, and samples with zero identifiable arthropod prey items were excluded. We conducted data reduction using Principal Component Analysis (PCA) with the prcomp function in R. For explanatory variables, we considered the number of prey items in each order, but not the number of rocks or plants. We compared *I. alvarius* diet data between urbanized and native habitats and *I. alvarius* diet data to those of sympatric anuran species (pooled due to low sample sizes of each sympatric species individually). We visualized PCAs using the *ade4* package [47] and used the aov() function to test for group differences in principal components. To test whether numbers of hymenopterans, beetles, native arthropods, and introduced arthropods varied between *I. alvarius* in native and urbanized habitats, we used Wilcoxon rank sum tests with continuity correction (wilcox.test in the R base package). We also used Wilcoxon rank sum tests to test whether the number of beetles, hymenopterans, and other arthropods (pooled) varied between *I. alvarius* and other sympatric anurans. We considered adjusting for anuran weight or age class in our analyses, but we did not detect a significant relationship between these variables and diet composition in exploratory generalized linear models. We performed data visualization initially in R using the *ggplot2* package [48] and then added visual elements to figures using Adobe Illustrator and BioRender.

## Results

### 5-MeO-DMT detected in all *I. alvarius* secretions

We confirmed that *I. alvarius* secretions contain 5-MeO-DMT using Direct Analysis in Real Time Mass Spectrometry (DART-TOF MS; Figs 1 and S1). Indeed, 5-MeO-DMT was detected in parotoid and leg gland secretions from all 26 *I. alvarius* individuals. We also recorded the presence of 5-HO-DMT (bufotenine), 5-MeO-N-methyltryptamine, MeO-tryptamine, 5-MeO-tryptophol and tryptophan in the secretions of *I. alvarius*. Bufotenine, 5-MeO-N-methyltryptamine, MeO-tryptamine, 5-MeO-tryptophol and tryptophan were also detected in *A. cognatus* or *A. punctatus* (Figs 1 and S1 and S3 Table). We detected more diversity in tryptamine-derived compounds in *I. alvarius* ventral and dorsal samples than secretions from the parotoid glands (S3 Table).

### *Incilius alvarius* are dietary generalists

We classified 216 arthropod prey items to seven orders from flushed stomach content and fecal samples of *I. alvarius*. The dominant prey types were hymenopterans and coleopterans (beetles). We found 135 hymenopterans (62.5% of total prey items) in the stomachs of nine *I. alvarius* (33%), and 68 beetles (31.5% of total prey items) in the stomachs of 17 *I. alvarius* (56.7%). Of the 135 hymenopterans, 134 were ants. We also found small numbers of Orthoptera, Scorpiones, Blattodea, Araneae and Odonata (Fig 2; S1 Table).

We further classified 185 prey items to 12 families, 39 prey items to 18 genera, and 24 prey items to 10 species in *I. alvarius* stomach contents and fecal samples. A full list of identified taxa is available in S1 and S4 Tables. Among the prey items, we identified a large wasp (Vespidae, Fig 3) and an Arizona bark scorpion (*Centruroides sculpturatus*, Fig 3) in the stomach contents of two different adult *I. alvarius*. Both species are toxic and capable of delivering formidable stings [49,50].

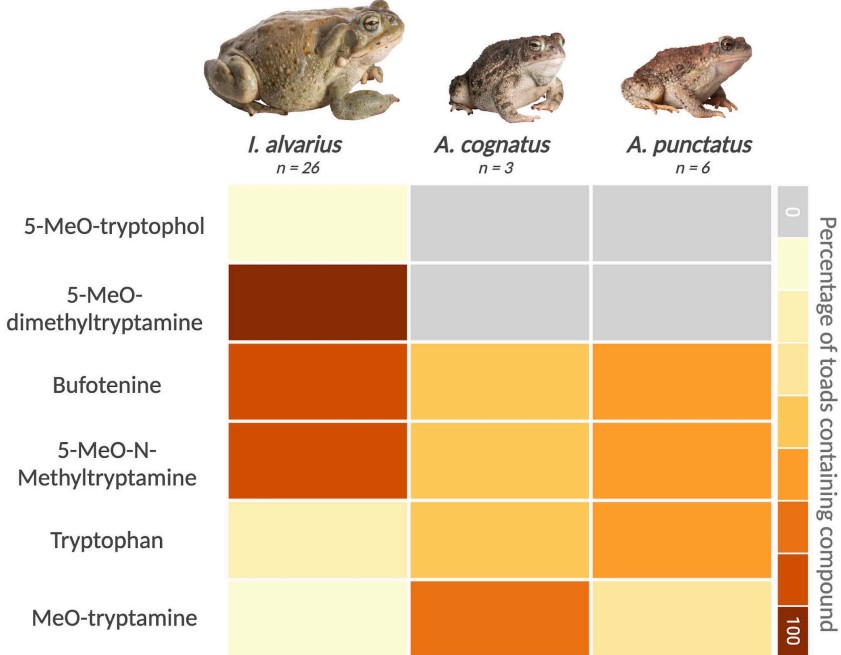

**Fig 1. Tryptophan derivatives of interest found in *Anaxyrus cognatus*, *Anaxyrus punctatus* and *Incilius alvarius*.** A heatmap indicates the fraction of toads in each species (rows) for which a given compound was detected (columns).

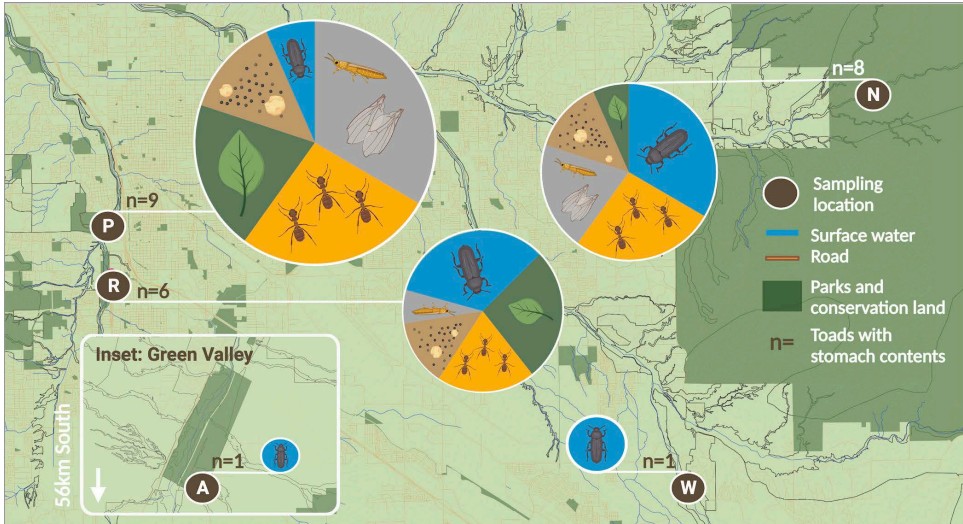

**Fig 2. Diet Composition of *Incilius alvarius* from 5 field sites in southeastern Arizona.** Pie charts represent the average diet composition for all toads at a given site. Diet items are coded in blue = beetles; orange = ants; gray = other arthropods; brown = rocks and sand; green = plant material. Site abbreviations correspond to Table 1, and locations are approximate due to conservation concerns (see Table 1). The baselayer for this diagram was designed Marina Luccioni in QGIS by custom super-imposing and re-coloring map data (street network, topography, riparian classification, parks and recreational areas, surface water features, conservation land categories) from publicly available GIS Open Data Pima County Maps (https://gisopendata.pima.gov/), published under a CC-BY license. Pie charts and insect diagrams were compiled with BioRender.com.

In addition to arthropod prey, we identified plant matter and inorganic material in the stomach contents of many *I. alvarius*. Of the 30 *I. alvarius* sampled, 13 (43%) had plant matter and 7 (23%) had rocks or sand (gastroliths) in their stomach or fecal contents. Specific items included leaves, stems, burs, cactus spines, and a bullet casing [51].

### *Incilius alvarius* in native and urban habitats differ in diet

We first explored the overall patterns of diet in *I. alvarius* using a principal component analysis (PCA) of prey grouped by order (Fig 4). Principal component (PC) 1 explained 26.5% of the variance while PC2 explained 20.5%. Toads from different habitats separated significantly by diet in PC2 ($F_1 = 6.21$, p = 0.02). We next tested whether specific classes of prey items differed in toads from native and urbanized habitats, but no significant differences were observed in the number of hymenopterans (Wilcoxon, W = 54.5, p = 0.48), beetles (Wilcoxon, W = 92.5, p = 0.09), or other arthropods pooled (Wilcoxon, W = 40, p = 0.08). We identified far more native arthropods than introduced arthropods in *I. alvarius* stomach content and fecal samples (211 native, 5 introduced). The five introduced arthropods were only found in toads from urban habitats and comprised cockroaches of two species (*Periplaneta americana* and *Shelfordella lateralis*).

### *Incilius alvarius* diet is broadly similar to that of sympatric anurans

To explore whether the *I. alvarius* exhibits dietary specialization, we also sampled the diet of three sympatric anurans that do not produce 5-MeO-DMT for comparison. We classified 64 prey items to 8 orders from flushed stomach contents and fecal samples of 7 *A. punctatus*, 4 *A. cognatus*, and 10 *S. couchii* (see S1 and S4 Tables and S1 Text for details). We first explored the overall patterns of diet using a PCA of prey grouped by order (Fig 5). Principal component (PC) 1 explained 23.1% of the variance while PC2 explained 18.6%, although species differences did not explain this variance (p ≥ 0.68 for both principal components). At the level of order, we documented similar diet composition between sympatric anurans and *I. alvarius*, where there were no significant differences in the number of hymenopterans (Wilcoxon, W = 136, p = 0.41),

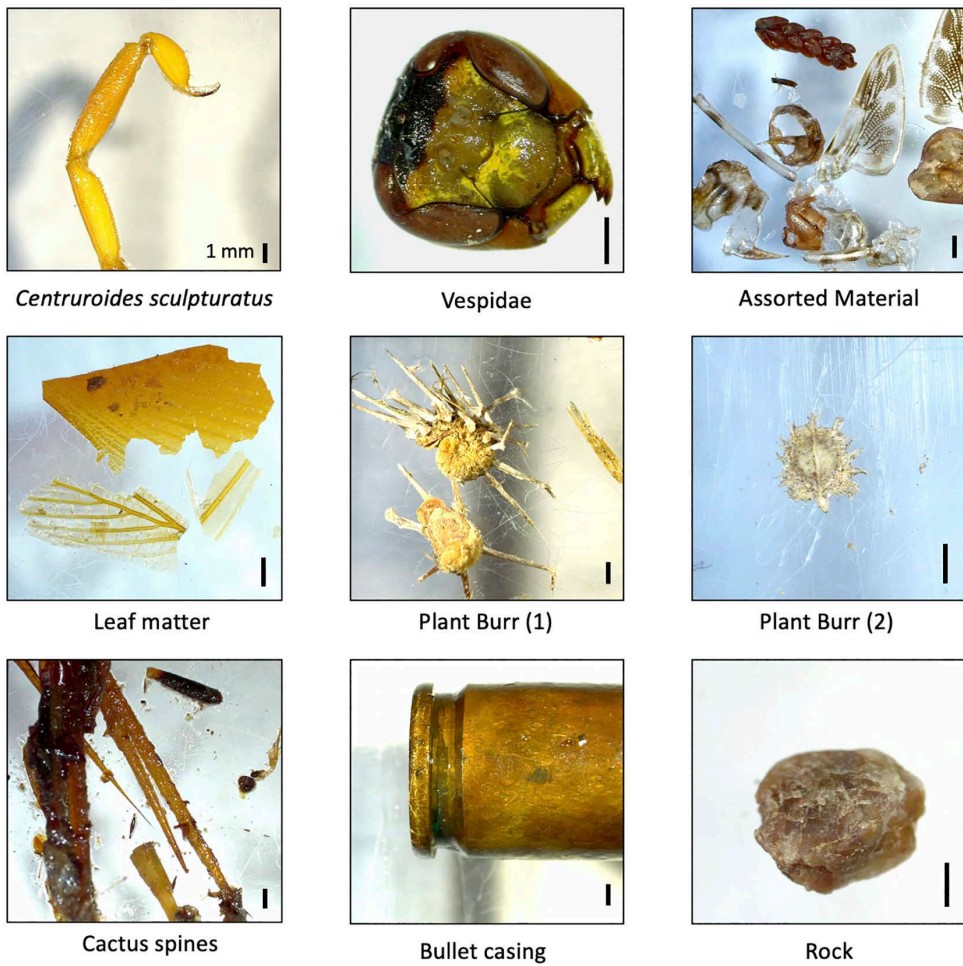

**Fig 3. Notable items from *Incilius alvarius* stomach and fecal contents.** All scale bars are 1 mm. Assorted material includes plant matter as well as wings and exoskeleton parts from a dragonfly, perhaps *Pantala flavescens*. The presence of a bullet casing in *Incilius alvarius* fecal contents is detailed in [51].

beetles (Wilcoxon, W = 212.5, p = 0.10) or other arthropods pooled (Wilcoxon, W = 164.5, p = 0.91) eaten. Overall, our data support the role of *I. alvarius* and sympatric anurans as dietary generalists and opportunists that consume a wide variety of prey.

## Discussion

### Direct dietary source of 5-MeO-DMT is unlikely in *I. alvarius*

Animals that sequester chemical defenses tend to demonstrate a preference for toxic prey compared to their non-sequestering relatives [52–54]. We predicted that if *I. alvarius* obtains 5-MeO-DMT through dietary sequestration, we would observe dietary differences between *I. alvarius* and sympatric anurans. Here, we found ants and beetles, many of which possess chemical defenses, to be the dominant prey items for *I. alvarius*. However, we found these groups to be dominant prey items for sympatric anurans as well. Overall, our results are in line with previous dietary studies of this species as well as many other bufonids [19,55]. However, we note that sampling stomach contents and fecal samples

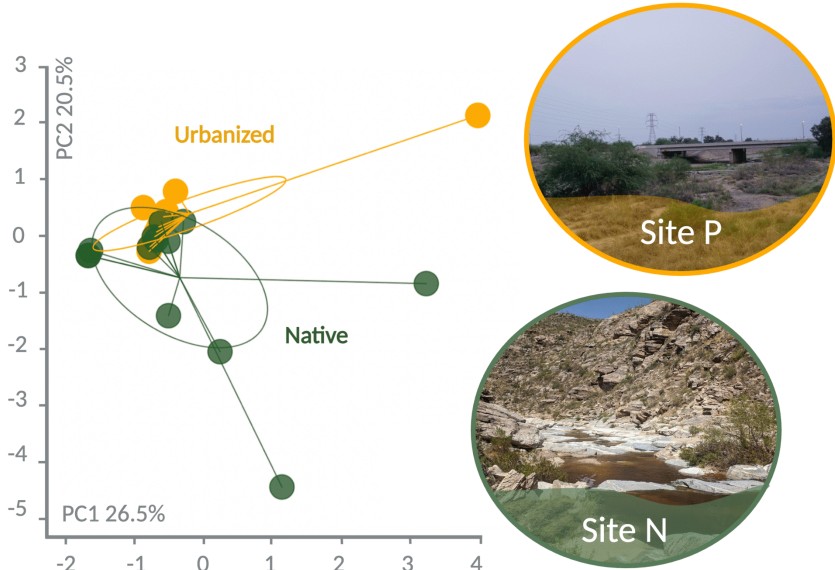

**Fig 4. Principal component analysis clustering of *Incilius alvarius* diet data (grouped by order), colored by habitat type (native green, urbanized orange).** Each point represents an individual toad; inertia ellipses graphically represent point clouds for each habitat. Photographs depict representative habitat from two of our five study sites classified as urbanized or native.

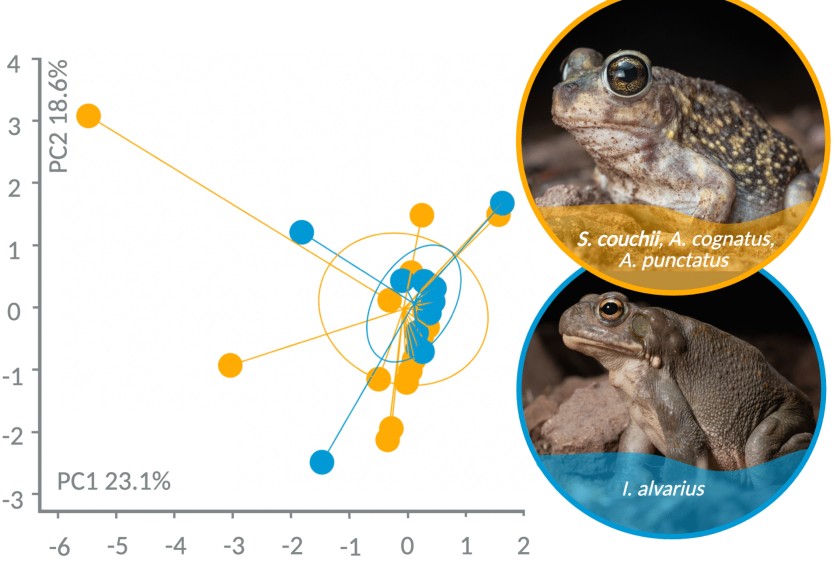

**Fig 5. Principal component analysis clustering of anuran diet data (grouped by order), colored by taxon (*Incilius alvarius* vs. other).** Each point represents an individual toad; inertia ellipses graphically represent point clouds for each taxon.

may overrepresent hard-shelled prey items, which are more likely to remain intact after consumption. Nevertheless, we documented such large numbers of ants and beetles that these two groups likely represent the dominant prey items for *I. alvarius*.

Many beetles and ants are chemically defended. *Incilius alvarius* was the only species in our study that we documented consuming *Eleodes* spp. darkling beetles (7 beetles across 3 toads). Although *Eleodes* produce benzoquinones [56,57] (also see Notable Prey below), there is no clear link to production of 5-MeO-DMT or immediate precursors in these taxa, and they may be avoided by (smaller) sympatric anurans simply due to their large size. We also note that several ants of the *Pogonomyrmex* genus were present in our sample set. Indigenous groups including the Kitanemuk, Kawaiisu, Tubatulabal, Interior Chumash, Yawelmani, Wikchamni, Yawdanchi, Bokninwad, Yokod, Palewyami, and possibly the Northern Miwok in California have knowledge and traditions surrounding the ingestion of harvester ants, likely *Pogonomyrmex californicus*, to induce hallucinations [58,59]. As the symptoms of ingesting *P. californicus* broadly match those of 5-MeO-DMT intoxication from *I. alvarius*, we initially speculated that *I. alvarius* could sequester 5-MeO-DMT from *Pogonomyrmex*. We found three *Pogonomyrmex* cf. *rugosus* (*P. rugosus* and *P. barbatus* hybridize extensively in the region [60], and our molecular identification methods produced variable results) in the stomach contents of two *I. alvarius*. However, we also found several *Pogonomyrmex* cf. *rugosus* in the stomach contents of two *S. couchii* and one *A. cognatus*. *Incilius alvarius* do not appear to specialize on these ants relative to their congeners, nor is there evidence that they eat them in sufficient quantities to provide a major source of 5-MeO-DMT.

Our diet results suggest that a direct dietary source of 5-MeO-DMT is unlikely in *I. alvarius*. This is reflected by a lack of prey specialization compared to sympatric species. Additionally, our principal component analysis and documentation of introduced prey in urban habitats only suggest that *I. alvarius* eat somewhat different prey in different habitats, likely reflecting differences in prey availability, but do not demonstrate corresponding differences in 5-MeO-DMT production. To further confirm these results, laboratory studies could vary the diet of captive toads and measure subsequent 5-MeO-DMT production, and/or use mass spectrometry to screen for 5-MeO-DMT in prey items. Repeated sampling of individuals may also be necessary to more fully catalog diet, as our stomach content samples represent a single snapshot in time for each individual.

## Alternatives to dietary sequestration

Several other possibilities exist for *I. alvarius* production or acquisition of 5-MeO-DMT. Erspamer et al. [17] concluded that 5-MeO-DMT is likely formed on the skin of Sonoran Desert toads via the O-methylation of bufotenine (5-HO-DMT), which represents a major component of the secretions from multiple bufonid toads including *I. alvarius*. Lerer et al. [61] recently found that *I. alvarius* cells cultured from parotoid gland tissue produce 5-MeO-DMT, suggesting that 5-MeO-DMT need not be exclusively diet-derived and can be produced from the basic molecular building blocks supplied in cell culture. Furthermore, Chen et al. [62] recently documented a different methyltransferase, an effective indolethylamine n-methyltransferase, in the related cane toad (*Rhinella marina*), a similarly sized bufonid toad native to the Neotropics that secretes bufotenine and other indolealkylamines, but not 5-MeO-DMT [63]. In light of our results, and those of these recent studies, published after completion of the current study, we consider *in vivo* transformation of 5-MeO-DMT from bufotenine a likely production pathway. Future research should focus on identifying and isolating the hypothesized 5-hydroxyindole-O-methyltransferase necessary for such a transformation.

The possibility of a plant or fungal source for 5-MeO-DMT should not be ruled out because the best-known sources of 5-MeO-DMT are from plants [22] and multiple other psychedelics are produced by fungi [64]. Many *I. alvarius* in our study ate plants. In previous literature these occurrences have been attributed to accidental ingestion, but this assumption may represent a "vegetal blind spot" in the diet literature. For example, a relative of *I. alvarius*, *Rhinella arenarum*, has been suspected to behave herbivorously during their dormant period, at least in laboratory conditions [65]. Many of our study anurans had plant matter in their stomach contents and/or items that were too homogenized or fragmented to identify using visual or Sanger sequencing methods but are likely identifiable with next-generation sequencing. Metabarcoding approaches with a focus on arthropod, plant and fungus-specific primers would help provide a more thorough estimation of diet composition for all the anurans in our dataset.

Finally, the skin microbiome may play a role in producing 5-MeO-DMT. Hayes et al. [66] document the biotransformation of *R. marina* bufadienolides by several bacteria species isolated from parotoid glands and Vaelli et al. [12] demonstrate the production of tetrodotoxin by bacterial symbionts on the skin of rough-skinned newts (*Taricha granulosa*). Although there is no published research on the microbiome of *I. alvarius*, the roles of these bacteria in related species raise the question of whether skin-dwelling microbial symbionts influence the composition of *I. alvarius* secondary metabolites.

## Other notable findings: Tryptophan derivatives in the skin and glands of *I. alvarius* and sympatric bufonids

Toad gland secretions contain a diverse cocktail of secondary metabolites, including various derivatives of tryptophan. In a recent study, Schwelm et al. [30] explored the toxin composition of five field-collected *I. alvarius* and noted 5-MeO-tryptamine and two positional isomers of hydroxylated MeO-DMT. We additionally recorded, for the first time as far as we are aware, the presence of 5-MeO-tryptamine in *A. cognatus* and 5-MeO-N-methyltryptamine in both *A. cognatus* and *A. punctatus* secretions. We recorded low amounts of 5-MeO-tryptamine and bufotenine for *I. alvarius* and it is likely that the high concentrations of 5-MeO-DMT in remaining samples inhibited detection of other molecules present at lower concentrations. This is supported by our findings that toxin extracts from the ventrum and dorsum, which contain less 5-MeO-DMT, possibly allowed for the detection of more diverse compounds. This inference is supported by the interpretation of the mass spectra of *I. alvarius*, where the high levels 5-MeO-DMT (i.e., the base peak) concealed the compounds below the threshold level of 5%. As Direct Analysis in Real Time (DART) ionization has limited quantitative capabilities, and because we ran variable amounts of toxin from each toad, we were not able to precisely quantify the percent abundance of 5-MeO-DMT in our samples. Thus, we were not able to perform a robust quantitative analysis of variation in 5-MeO-DMT production between toads. In future work, more quantitative approaches should be used to measure 5-MeO-DMT in Sonoran Desert toad secretions.

Bufotenine is a tryptamine alkaloid and an isomer of the psychedelic compound psilocin. We recorded the presence of bufotenine in *I. alvarius* and their congeners. Bufotenine was present in *A. punctatus* at levels >40% but only found in trace amounts (<5%) in *I. alvarius*, which is what we might expect to see if bufotenine is being used as a substrate for creating 5-MeO-DMT. Many additional tryptamines and derivatives reported in *I. alvarius* secretions [30,37–39] were also found in secretions of *A. cognatus and A. punctatus*. Some of these molecules are potential chemical precursors for 5-MeO-DMT. Understanding the similarities in toxin composition between *I. alvarius* and related anurans that likely do not produce 5-MeO-DMT can help pinpoint the ways that *I. alvarius* are unique and provide clues as to which pathways are used to create 5-MeO-DMT.

## Other notable findings: *Incilius alvarius* are dietary generalists capable of ingesting toxic and barbed prey

Our findings support previous results [19,32] that indicate *I. alvarius* are largely dietary generalists and opportunists, observed to consume a wide variety of prey, including arthropods with stings or toxic secretions [67]. However, they may exhibit some specialization on ants and beetles, as these two groups made up the vast majority of dietary prey items, which was also documented by Cole [19]. In addition to many ants and beetles, we also found spiders, cockroaches, dragonfly larvae, a grasshopper, a cricket, a wasp, and a scorpion in *I. alvarius* stomach contents. Unlike previous studies, we did not document vertebrate prey in the stomach contents of any *I. alvarius* sampled.

In line with Cole [19], we recorded a variety of arthropods with stings or toxic secretions consumed by *I. alvarius*. These included putative rough harvester ants (*Pogonomyrmex* cf. *rugosus*) and a vespid wasp, both of which readily sting and possess potent venoms [50,68], as well as a variety of *Eleodes* and other tenebrionid beetles, many of which produce a noxious secretion containing benzoquinones when disturbed [56,57]. Perhaps most remarkable was the tail of an Arizona bark scorpion (*Centruroides sculpturatus*), whose sting is extremely painful and potentially lethal to humans [49] and produces neurotoxic effects in the nerve fibers of at least one species of frog [69]. It is unknown how *I. alvarius* and their

relatives (we also found *Pogonomyrmex* in *A. cognatus* and *S. couchii* stomach contents) manage to consume arthropods with such formidable chemical defenses. Horned lizards (*Phrynosoma* spp.) have been shown to detoxify *Pogonomyrmex* ant venom via a blood factor [70], while southern grasshopper mice (*Onychomys torridus*) are resistant to bark scorpion venom in part due to a mutation in a non-target sodium channel [71]. It is possible that *I. alvarius* and other anurans employ similar mechanisms.

In addition to arthropod prey, we identified plant matter and inorganic material in the stomach contents of many *I. alvarius*. These findings are usually explained as the result of incidental capture, either as a byproduct when attempting to capture something else or the result of misidentification. However, gastroliths (stomach stones) are believed to aid in the mechanical breakdown of organic matter in the gut of multiple other taxa [72], raising the possibility that *I. alvarius* consume hard inorganic material for digestive purposes.

## Conclusion

We found limited evidence to support the hypothesis that *Incilius alvarius* 5-MeO-DMT production is linked to diet. Instead, we found that *I. alvarius* from multiple sites and habitats all produce high quantities of 5-MeO-DMT, despite a diversity of prey in their stomach and fecal contents. Furthermore, we found that *I. alvarius* have a largely similar diet to three sympatric anuran species which likely do not produce 5-MeO-DMT. Although our study was limited to single diet items, our results do not support a direct dietary arthropod source of 5-MeO-DMT in *I. alvarius*. Alternatively, these toads could produce 5-MeO-DMT endogenously using a 5-hydroxyindole-O-methyltransferase or via a bacterial symbiont, as previously hypothesized, or sequester 5-MeO-DMT from a plant or fungal source.

## Supporting information

**S1 Table. Summary of dietary items and flushed and fecal Samples from *I. alvarius* and sympatric species.** (XLSX)

**S2 Table. Description of samples for chemical analysis.** (DOCX)

**S3 Table. Tryptophan and derivatives from the parotoid glands and skin of sampled anurans.** (DOCX)

**S4 Table. Identifications of arthropod prey consumed by *I. alvarius* and sympatric species.** Includes all prey items identified to family or beyond, except ants, which comprised all hymenopteran specimens but one. (XLSX)

**S1 Fig. Presence and absence of various molecules across species sampled, as determined by Direct Analysis in Real Time Mass Spectrometry (DART-TOF MS) in the negative ion mode.** (TIF)

**S1 Text. Additional description of diet data for sympatric anurans.** (DOCX)

## Acknowledgments

We are grateful to a wonderful team who made this study possible: Leisy and Michael Wyman allowed us facilities access and indispensable fieldwork support, Robert Villa shared his expertise regarding wide-ranging aspects of the study and provided a useful review of our manuscript, John Wiens at the University of Arizona stored our samples from August to September 2020, Nora Martin advised our stomach flushing protocol, Dave Ramirez provided valuable

logistical support and Katie Fiocca shared helpful comments on our manuscript. Many people generously assisted with visual identification of invertebrates from stomach and fecal samples, including Brad Barnd (BugGuide), Michael Bogan, David Donoso, Curt Harden (BugGuide), Solomon Hendrix (BugGuide), Andrew Johnston, Blaine Mathison (BugGuide) and Wyatt Mendez. We also thank Alberto Arevalo and Elizabeth Navarro for their support in creating photographs for select figures, and Quinn Agnew, Daniel Carhuff, Andrew Connoy and Tasman Ezra for their assistance with our fieldwork.

## Author contributions

**Conceptualization:** Marina D. Luccioni, Jules T. Wyman, Lauren A. O'Connell.

**Data curation:** Marina D. Luccioni, Jules T. Wyman.

**Formal analysis:** Marina D. Luccioni, Jules T. Wyman, Edgard O. Espinoza, Lauren A. O'Connell.

**Funding acquisition:** Marina D. Luccioni, Lauren A. O'Connell.

**Investigation:** Marina D. Luccioni, Jules T. Wyman, Edgard O. Espinoza, Lauren A. O'Connell.

**Methodology:** Marina D. Luccioni, Jules T. Wyman, Edgard O. Espinoza, Lauren A. O'Connell.

**Project administration:** Marina D. Luccioni, Jules T. Wyman, Lauren A. O'Connell.

**Resources:** Marina D. Luccioni, Lauren A. O'Connell.

**Software:** Marina D. Luccioni, Jules T. Wyman.

**Supervision:** Marina D. Luccioni, Jules T. Wyman, Lauren A. O'Connell.

**Validation:** Marina D. Luccioni, Jules T. Wyman.

**Visualization:** Marina D. Luccioni, Jules T. Wyman, Edgard O. Espinoza.

**Writing – original draft:** Marina D. Luccioni, Jules T. Wyman, Edgard O. Espinoza, Lauren A. O'Connell.

**Writing – review & editing:** Marina D. Luccioni, Jules T. Wyman, Edgard O. Espinoza, Lauren A. O'Connell.

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
