## [Decision Letter · Decision Letter 0]

9 Apr 2025

Dear Dr. Wyman,

Thank you for submitting your manuscript to PLOS ONE. After careful consideration, we feel that it has merit but does not fully meet PLOS ONE’s publication criteria as it currently stands. Therefore, we invite you to submit a revised version of the manuscript that addresses the points raised during the review process.

https://journals.plos.org/plosone/s/submission-guidelines#loc-laboratory-protocols . Additionally, PLOS ONE offers an option for publishing peer-reviewed Lab Protocol articles, which describe protocols hosted on protocols.io. Read more information on sharing protocols at https://plos.org/protocols?utm_medium=editorial-email&utm_source=authorletters&utm_campaign=protocols .

We look forward to receiving your revised manuscript.

Kind regards,

Natan Maciel

Academic Editor

PLOS ONE

Journal Requirements:

3. To comply with PLOS ONE submissions requirements, in your Methods section, please provide additional information regarding the experiments involving animals and ensure you have included details on (1) methods of sacrifice, (2) methods of anesthesia and/or analgesia, and (3) efforts to alleviate suffering.

“We are grateful to a wonderful team who made this study possible: Leisy and Michael Wyman allowed us facilities access and indispensable fieldwork support, Robert Villa shared his expertise regarding wide-ranging aspects of the study and provided a useful review of our manuscript, John Weins at the University of Arizona stored our samples from August to September 2020, Nora Martin advised our stomach flushing protocol, Dave Ramirez provided valuable logistical support and Katie Fiocca shared helpful comments on our manuscript. Many people generously assisted with visual identification of invertebrates from stomach and fecal samples, including Brad Barnd (BugGuide), Michael Bogan, David Donoso, Curt Harden (BugGuide), Solomon Hendrix (BugGuide), Andrew Johnston, Blaine Mathison (BugGuide) and Wyatt Mendez. We also thank Alberto Arevalo and Elizabeth Navarro for their support in creating photographs for select figures, and Quinn Agnew, Daniel Carhuff, Andrew Connoy and Tasman Rosenfeld for their assistance with our fieldwork. This work was funded by a New York Stem Cell Foundation to LAO. LAO is a New York Stem Cell Foundation – Robertson Investigator.”

“LAO received funding from the New York Stem Cell Foundation (https://nyscf.org/) for this work. LAO is a New York Stem Cell Foundation – Robertson Investigator. The New York Stem Cell Foundation did not play any role in the study design, data collection and analysis, decision to publish, or preparation of this manuscript.”

6. We note that Figures 1, 2, 3, 4, 5 and S1 in your submission contain copyrighted images. All PLOS content is published under the Creative Commons Attribution License (CC BY 4.0), which means that the manuscript, images, and Supporting Information files will be freely available online, and any third party is permitted to access, download, copy, distribute, and use these materials in any way, even commercially, with proper attribution. For more information, see our copyright guidelines: http://journals.plos.org/plosone/s/licenses-and-copyright.

a. You may seek permission from the original copyright holder of Figures 1, 2, 3, 4, 5 and S1 to publish the content specifically under the CC BY 4.0 license.

7. (http://creativecommons.org/licenses/by/4.0/). Natural Earth (public domain): http://www.naturalearthdata.com/ We note that Figure 2 in your submission contain map/satellit] image which may be copyrighted. All PLOS content is published under the Creative Commons Attribution License (CC BY 4.0), which means that the manuscript, images, and Supporting Information files will be freely available online, and any third party is permitted to access, download, copy, distribute, and use these materials in any way, even commercially, with proper attribution. For these reasons, we cannot publish previously copyrighted maps or satellite images created using proprietary data, such as Google software (Google Maps, Street View, and Earth). For more information, see our copyright guidelines: http://journals.plos.org/plosone/s/licenses-and-copyright.

Additional Editor Comments:

Dear Dr. Wyman,

Firstly, my apologies for the long delay to reach a decision regarding your manuscript. This delay was due to changes of editor and issues to find available reviewers. To our knowledge, your work represents the first to test whether the toad I. alvarius 5-MeO-DMT secretion is related to a dietary source, together with an interesting diet study.

I have received the reviewer's comments on your manuscript. Based on their evaluation your manuscript is considered accepted upon minor revision. Both reviewers believe that your manuscript is nice and well written and I agree. Although not completely clear due some kind of issue in the Plos platform (please see above), the Reviewer 1 seems to believe that diet is not probably the most likely the answer to the uniqueness of I. alvarius 5-MeO-DMT presence, but more a hypothesis based on bacterial symbiosis or a hydroxylase presence in the toad skin. The reviewer 2 have a similar opinion regarding the origin of 5-MeO-DMT in I. alvarius still not clarified. Perhaps the manuscript could be focus more on testing the hypothesis that the origin of 5-MeO-DMT in I. alvarius is not related do dietary origin rather than testing that the hypothesis that is related. Is a fine line, but it changes the manuscript presentation. Please let me know your thoughts regarding this point.

Please see remaining comments and corrections of the reviewers below. I am looking forward to receive a revised version of your manuscript including replies to the essential critiques and comments of the reviewers through the online editorial system. For those to which you disagree a justification must be presented.Thank you for submitting your work to PlosOne!  If you have any further questions please let me know. Natan Maciel.

Reviewer 1

This is a well written manuscript that reports the dietary regime of desert toads in an effort to determine if a potent compound 5-Me0--DMT has a dietary source. This reviewer would be more comfortable if there were less effort to sell the idea that these toads are special and

D-24-26523

Abstract

Fine

Intro—Why is work on what used to be called Bufo marinas and other bufonids that have compounds related 5-Me0-DMT not included?

85-86 delete. Does not add to story.

Methods

122- why weren’t samples “suitable”?

Results

Intro and Discussion

So three additional species contain 5-Me0-tryptamine or a close relative?

That leaves the spadefoot for diet comparison?

Why not look for the hydroxylase?

Reviewer 2

The study aims to investigate the origin of the compound 5-MeO-DMT, present in high levels in the skin of the Sonoran toad Incilius alvarius. Although the existence of this compound in this species has been known for a long time, its origin remains unknown. The authors used a comparative ecological, biochemical and molecular biology approach, analyzing the skin secretions and the diet of I. alvarius and three other sympatric toad species from 5 locations with distinctive ecological characteristics. The analysis of this data set showed that the prey consumed by all species is very similar, and all lack 5-MeO-DMT. Thus, the explanation for the existence of the compound solely in I. alvarius must be based on endogenous production or originating from the bacterial flora of the skin of these animals, as was recently demonstrated in the salamander Taricha granulosa, whose skin is very rich in the toxin TTX. The work is well written, well-illustrated, and methodologically appropriate, and it frankly discusses the points that still need further clarification. However, it sheds light on an open question that may pave the way for new investigations capable of better clarifying the actual origin of 5-MeO-DMT in I. alvarius.

Below are a few comments on points I observed while reading the text.

Line 47 – Taricha granulosa??

Line 57 – I believe “agitated” is not the best term here. Maybe “theatened” or “attacked”??

Table 1, legend - Sc = Scaphiopus couchii (and not Sceloporus couchii)

Line 132 – “Secretions were collected in the field using kimwipes”. I understand that the authors are referring here to the secretion coming from the parotoids. Did the animals naturally release the skin secretion, or were they mechanically stimulated to secrete? This point could be better explained.

Reviewers' comments:

Reviewer's Responses to Questions

**Comments to the Author**

1. Is the manuscript technically sound, and do the data support the conclusions?

Reviewer #1: Yes

Reviewer #2: Yes

2. Has the statistical analysis been performed appropriately and rigorously?

Reviewer #1: Yes

Reviewer #2: Yes

3. Have the authors made all data underlying the findings in their manuscript fully available?

Reviewer #1: Yes

Reviewer #2: Yes

4. Is the manuscript presented in an intelligible fashion and written in standard English?

Reviewer #1: Yes

Reviewer #2: Yes

Reviewer #1: This is a well written manuscript that reports the dietary regime of desert toads in an effort to determine if a potent compound 5-Me0--DMT has a dietary source. This reviewer would be more comfortable if there were less effort to sell the idea that these toads are special and

D-24-26523

Abstract

Fine

Intro—Why is work on what used to be called Bufo marinas and other bufonids that have compounds related 5-Me0-DMT not included?

85-86 delete. Does not add to story.

Methods

122- why weren’t samples “suitable”?

Results

Intro and Discussion

So three additional species contain 5-Me0-tryptamine or a close relative?

That leaves the spadefoot for diet comparison?

Why not look for the hydroxylase?

Reviewer #2: The study aims to investigate the origin of the compound 5-MeO-DMT, present in high levels in the skin of the Sonoran toad Incilius alvarius. Although the existence of this compound in this species has been known for a long time, its origin remains unknown. The authors used a comparative ecological, biochemical and molecular biology approach, analyzing the skin secretions and the diet of I. alvarius and three other sympatric toad species from 5 locations with distinctive ecological characteristics. The analysis of this data set showed that the prey consumed by all species is very similar, and all lack 5-MeO-DMT. Thus, the explanation for the existence of the compound solely in I. alvarius must be based on endogenous production or originating from the bacterial flora of the skin of these animals, as was recently demonstrated in the salamander Taricha granulosa, whose skin is very rich in the toxin TTX. The work is well written, well-illustrated, and methodologically appropriate, and it frankly discusses the points that still need further clarification. However, it sheds light on an open question that may pave the way for new investigations capable of better clarifying the actual origin of 5-MeO-DMT in I. alvarius.

Below are a few comments on points I observed while reading the text.

Line 47 – Taricha granulosa??

Line 57 – I believe “agitated” is not the best term here. Maybe “theatened” or “attacked”??

Table 1, legend - Sc = Scaphiopus couchii (and not Sceloporus couchii)

Line 132 – “Secretions were collected in the field using kimwipes”. I understand that the authors are referring here to the secretion coming from the parotoids. Did the animals naturally release the skin secretion, or were they mechanically stimulated to secrete? This point could be better explained.

**Do you want your identity to be public for this peer review?** For information about this choice, including consent withdrawal, please see our Privacy Policy

Reviewer #1: No

Reviewer #2: No

---

## [Author Response · Author response to Decision Letter 1]

29 Aug 2025

August 28, 2025

Editorial Board of PLOS One

Response to Review Comments for Manuscript PONE-D-24-26523: Diet and chemical defenses of the Sonoran Desert toads

Response: Thank you very much for the detailed review of our manuscript and for the opportunity to make revisions. We are grateful for your feedback and that of the two anonymous referees. Our revised manuscript was edited accordingly. Below we respond in detail to all comments from yourself and both reviewers.

Response: We appreciate your providing these templates, and have now ensured that our manuscript meets all style requirements of the journal.

Response: All work was covered under Arizona Game and Fish Department (AZGFD) scientific collecting licenses. All field sites were on public land, approved by AZGFD for sampling, and specifically covered under our licenses. We have updated our Methods section to clarify this (see lines 136-139).

3. To comply with PLOS ONE submissions requirements, in your Methods section, please provide additional information regarding the experiments involving animals and ensure you have included details on (1) methods of sacrifice, (2) methods of anesthesia and/or analgesia, and (3) efforts to alleviate suffering.

Response: The requested additional information for (1-3) is now provided in lines 139-143.

“We are grateful to a wonderful team who made this study possible: Leisy and Michael Wyman allowed us facilities access and indispensable fieldwork support, Robert Villa shared his expertise regarding wide-ranging aspects of the study and provided a useful review of our manuscript, John Weins at the University of Arizona stored our samples from August to September 2020, Nora Martin advised our stomach flushing protocol, Dave Ramirez provided valuable logistical support and Katie Fiocca shared helpful comments on our manuscript. Many people generously assisted with visual identification of invertebrates from stomach and fecal samples, including Brad Barnd (BugGuide), Michael Bogan, David Donoso, Curt Harden (BugGuide), Solomon Hendrix (BugGuide), Andrew Johnston, Blaine Mathison (BugGuide) and Wyatt Mendez. We also thank Alberto Arevalo and Elizabeth Navarro for their support in creating photographs for select figures, and Quinn Agnew, Daniel Carhuff, Andrew Connoy and Tasman Rosenfeld for their assistance with our fieldwork. This work was funded by a New York Stem Cell Foundation to LAO. LAO is a New York Stem Cell Foundation – Robertson Investigator.”

“LAO received funding from the New York Stem Cell Foundation (https://nyscf.org/) for this work. LAO is a New York Stem Cell Foundation – Robertson Investigator. The New York Stem Cell Foundation did not play any role in the study design, data collection and analysis, decision to publish, or preparation of this manuscript.”

Response: Thank you for bringing this to our attention. We have removed the following line from our Acknowledgements: “This work was funded by a New York Stem Cell Foundation to LAO. LAO is a New York Stem Cell Foundation – Robertson Investigator.” We would like to maintain the Funding Statement in its current form.

Response: All data are now publicly available through Dryad and Genbank. Accession information is included in the updated version of our manuscript.

6. We note that Figures 1, 2, 3, 4, 5 and S1 in your submission contain copyrighted images. All PLOS content is published under the Creative Commons Attribution License (CC BY 4.0), which means that the manuscript, images, and Supporting Information files will be freely available online, and any third party is permitted to access, download, copy, distribute, and use these materials in any way, even commercially, with proper attribution. For more information, see our copyright guidelines: http://journals.plos.org/plosone/s/licenses-and-copyright.

a. You may seek permission from the original copyright holder of Figures 1, 2, 3, 4, 5 and S1 to publish the content specifically under the CC BY 4.0 license.

Response: All photographs included in the manuscript are the property of the authors.

7. We note that Figure 2 in your submission contain map/satellit] image which may be copyrighted. All PLOS content is published under the Creative Commons Attribution License (CC BY 4.0), which means that the manuscript, images, and Supporting Information files will be freely available online, and any third party is permitted to access, download, copy, distribute, and use these materials in any way, even commercially, with proper attribution. For these reasons, we cannot publish previously copyrighted maps or satellite images created using proprietary data, such as Google software (Google Maps, Street View, and Earth). For more information, see our copyright guidelines: http://journals.plos.org/plosone/s/licenses-and-copyright.

Response: Confirmation of a CC-BY license has been obtained for the map from Figure 2 which was obtained from publicly available GIS Open Data Pima County Maps (https://gisopendata.pima.gov/).

Response: Captions are now provided.

Response: We have reviewed our references list and ensured it is complete and correct. Changes from the original submission are as follows:

We rectified our citation of [33] (Brown, DE. Biotic communities: southwestern United States and northwestern Mexico. University of Utah Press; 1994.), which was originally cited in the text but not included in the References list.

We added a reference to our Materials and methods section [48] to clarify our data visualization methods.

We added two references to our Discussion section, [61] and [62]. These articles represent new findings since the conclusion of our study that are relevant to the interpretation of our results.

Where necessary, we made minor formatting corrections (e.g. renumbering, removing extra spacing) to our References section and in-text citations.

Additional Editor Comments:

Dear Dr. Wyman,

Firstly, my apologies for the long delay to reach a decision regarding your manuscript. This delay was due to changes of editor and issues to find available reviewers. To our knowledge, your work represents the first to test whether the toad I. alvarius 5-MeO-DMT secretion is related to a dietary source, together with an interesting diet study.

I have received the reviewer's comments on your manuscript. Based on their evaluation your manuscript is considered accepted upon minor revision. Both reviewers believe that your manuscript is nice and well written and I agree. Although not completely clear due some kind of issue in the Plos platform (please see above), the Reviewer 1 seems to believe that diet is not probably the most likely the answer to the uniqueness of I. alvarius 5-MeO-DMT presence, but more a hypothesis based on bacterial symbiosis or a hydroxylase presence in the toad skin. The reviewer 2 have a similar opinion regarding the origin of 5-MeO-DMT in I. alvarius still not clarified. Perhaps the manuscript could be focus more on testing the hypothesis that the origin of 5-MeO-DMT in I. alvarius is not related do dietary origin rather than testing that the hypothesis that is related. Is a fine line, but it changes the manuscript presentation. Please let me know your thoughts regarding this point.

Please see remaining comments and corrections of the reviewers below. I am looking forward to receive a revised version of your manuscript including replies to the essential critiques and comments of the reviewers through the online editorial system. For those to which you disagree a justification must be presented.Thank you for submitting your work to PlosOne! If you have any further questions please let me know. Natan Maciel.

Response: We are thrilled to hear of our acceptance pending minor revision. Thank you very much for your compliments about the manuscript, and all of your work finding reviewers and synthesizing and providing feedback. We have done our best to address all comments and concerns herein.

In regard to changing the presentation of our hypothesis, while we understand there is good reason to think that 5-MeO-DMT is produced endogenously by an enzyme, we feel that maintaining our hypothesis in its current form is the most honest presentation of our initial aim–investigating diet as a source for 5-MeO-DMT production. Based on the criteria identified by Savitsky et al. [26], outlined in our Introduction, I. alvarius is a candidate taxon for defensive sequestration, and we felt our hypothesis was worthy of investigation

---

## [Editor Report · Decision Letter 1]

14 Oct 2025

Diet and chemical defenses of the Sonoran Desert toads

PONE-D-24-26523R1

Dear Dr. Wyman,

We’re pleased to inform you that your manuscript has been judged scientifically suitable for publication and will be formally accepted for publication once it meets all outstanding technical requirements.

Kind regards,

Natan Maciel

Academic Editor

PLOS ONE

Additional Editor Comments (optional):

Dear Authors,

My sincere apologies for the long delay to reach a decision regarding your manuscript. I think you did a great work in order to incorporate the corrections and comments of the reviewers as well as presenting justification to those points to which you disagree from us. Also thank you for your effort to reach the additional requirements of Plos One such as information of license permits and copyrighted images. Thus, I believe you an improved and more fluid to read manuscript and I glad to considered it accepted!

When reading the revised version of your manuscript I noticed some tiny writing points that could be accommodated in the final version, if you accept. 1) Since the Sonoran desert toad is a species, the manuscript title could be: "Diet and chemical defenses of the Sonoran Desert toad" instead of "Diet and chemical defenses of the Sonoran Desert toads"; 2) It is common practice not to abbreviate the genus names at the beginning of sentences. Therefore, I suggest reviewing the text and correct it, including the titles of some sections of the manuscript, such as: "I. alvarius are dietary generalists"; 3) In Table 1, the species names are not italicized; 4) In Table S2, change "Alvarius" to "alvarius"; 5) In Table S3, some species names are not italicized.

Thank you for submitting your work to us! I'm looking forward to see the manuscript published. 

If you have any further questions please let me know. 

Natan Maciel.
---

## [Editor Report · Acceptance letter]

PONE-D-24-26523R1

PLOS ONE

Dear Dr. Wyman,

I'm pleased to inform you that your manuscript has been deemed suitable for publication in PLOS ONE. Congratulations! Your manuscript is now being handed over to our production team.

Kind regards,

on behalf of

Dr. Natan Maciel

Academic Editor

PLOS ONE